# FROM REST TO ACTION: ADAPTIVE WEIGHT GENERATION FOR MOTOR IMAGERY CLASSIFICATION FROM RESTING-STATE EEG USING HYPERNETWORKS

## ABSTRACT

Existing EEG-based brain-computer interface (BCI) systems require long calibration sessions from the intended users to train the models, limiting their use in real-world applications. Additionally, despite containing user-specific information and features correlating with BCI performance of a user, resting-state EEG data is underutilized, especially in motor imagery decoding tasks. To address the challenge of within and across-user generalisation, we propose a novel architecture, Hyper-EEGNet, which integrates HyperNetworks (HNs) with the EEGNet architecture to adaptively generate weights for motor imagery classification based on resting-state data. Our approach performs similarly in a Leave-Subject-Out scenario using a dataset with 9 participants, compared to the baseline EEGNet. When the dataset size is scaled, with 33 participants' datasets, the model demonstrates its generalisation capabilities using the information from resting state EEG data, particularly when faced with unseen subjects. Our model can learn robust representations in both cross-session and cross-user scenarios, opening a novel premise to leverage the resting state data for downstream tasks like motor imagery classification. The findings also demonstrate that such models with smaller footprints reduce memory and storage requirements for edge computing. The approach opens up avenues for faster user calibration and better feasibility of edge computing, a favourable combination to push forward the efforts to bring BCIs to real-world applications.

## 1 INTRODUCTION

The growing use of electroencephalograms (EEGs) in brain-computer interfaces (BCIs) has gained attention due to their non-invasive nature and high temporal resolution, making them ideal for decoding brain activity patterns in real-time (Schalk et al., 2024). BCIs, providing an interface between the brain and external devices, have applications in neurorehabilitation, assistive technologies, and neuroprosthetics. Among the various paradigms within BCIs, motor imagery (MI) decoding, which involves classifying imagined movements by the users from EEG signals, is of greater interest for decoding motor control. However, despite advances in hardware and software pipelines, MI-based BCIs have substantial challenges to bring them to real-world applications for generalised usage. The challenges with non-invasive BCIs are particularly in achieving robust and consistent performance across users and sessions (Saha & Baumert, 2020).

An outstanding challenge in MI-BCI systems is the variability in brain signals across users and sessions. This variability arises from differences in individual neural patterns, low signal-to-noise ratio, and varying conditions of the users, like fatigue or attention (Pan et al., 2022; Kobler et al., 2022). These differences cause inconsistencies in decoding MI patterns, limiting the generalizability of BCIs in real-world applications. The ability to generalize across users and sessions is necessary for practical and accessible applications of BCIs, especially in scenarios where collecting large amounts of personalized data is unfeasible.

In addition to cross-user variability, BCI performance is also hindered by BCI illiteracy (Allison & Neuper, 2010), where certain individuals cannot generate the neural signals necessary for BCI

control. Though the concept of BCI illiteracy has been debated and its cause is a subject of research (Becker et al., 2022; Thompson, 2019; Alonso-Valerdi & Mercado-García, 2021), the result is evident across datasets. This makes designing universally effective BCIs challenging. Research has also focused on the predictors of BCI performance, helping to pre-identify individuals who may face difficulties with BCIs, which may guide personalized interventions or optimizations. Importantly, much of the research has used resting-state EEG data to develop these predictors (Tzdaka et al., 2020; Trocellier et al., 2024; Blankertz et al., 2010). Resting-state data, collected while the user is relaxed, offers insights into baseline brain activity without task-specific requirements, making it an attractive candidate for predicting BCI proficiency. Moreover, resting state EEG data is also a marker for user identification (Ma, 2015; Choi et al., 2018; Wang et al., 2019), depicting user-specific information.

Data-driven deep-learning models have effectively improved BCI performance on different tasks (Hossain et al., 2023; Tibrewal et al., 2022). Transfer learning, where models trained on data from one individual or group can be adapted to another, holds promise for creating systems that work across different users, including able-bodied and SCI patients (Nagarajan et al., 2024; Xu et al., 2021). Furthermore, a study by Camille Benaroch & Lotte (2022) has used user-specific frequencies to optimize decoding algorithms, applying data-driven approaches. However, to our knowledge, this is the first work to use resting state EEG data to train the model for motor-imagery classification. The major contributions of this work are as follows:

- Propose a novel HyperEEGNet architecture using HyperNetworks to learn unique user-specific representations as adaptive weights for the underlying task in EEG decoding.
- Demonstrate the significance of resting state EEG data to solve downstream tasks like motor imagery classification using data-driven learning.

## 2 METHOD

### 2.1 DATASETS

The dataset used in this study consists of electroencephalogram (EEG) recordings from 87 individuals who participated in motor imagery (MI) tasks and resting-state conditions Dreyer et al. (2023). The dataset is unique, given the large number of participants and the available recordings for each user. The EEG data were collected using 27 electrodes placed with a 10-20 configuration system, each sampling at a rate of 512 Hz. The dataset consisted of 70 hours of recordings of 8-second long runs when participants performed motor imagery, i.e. imagining left and right-hand movements following a visual cue on the screen.

This work used the sub-dataset "A" with 60 participants. The dataset mentions that 18 participants reported having noisy channel data or distractions from the environment during the sessions. These participants are ignored in the study. The dataset has two runs for each participant, which were used for training the model, while the rest of the four runs are termed "online" runs. Following the benchmark set by Dreyer et al. Dreyer et al. (2023), each participant's two "acquisition" runs are used for training, and the four online runs are used as test sets for the within-user across-session scenario. For the across-user scenario, data from the last 9 participants ( 20%) was considered a test set, while the rest of the data from the 33 participants was used for training. A band-pass filter with a frequency range of 0.5-40 Hz was used to prepare the raw EEG data for analysis. Epochs, or time segments of EEG data, were created by segmenting the 3 seconds of data following the event marker at the onset of the visual cue for movement imagination. The resting state data was extracted from the first two seconds of the trial, where the participants focused on a fixation cue and were not explicitly instructed to rest.

To understand the effectiveness of the proposed approach on a comparatively smaller dataset with 9 participants, BNCI 2014 IIa Competition Dataset Brunner et al. (2008) is used. The dataset consists of electroencephalogram (EEG) signals from 9 individuals who participated in motor imagery (MI) tasks and resting state conditions. The EEG data were collected using 22 electrodes, each sampling at a frequency of 250 Hz. The analysis involved two classes: right-hand and left-hand movement imagery, while feet and tongue movements were ignored. Each epoch consisted of 4 second-long motor imagery activity. The resting state data was extracted from the first two seconds of the trial, where the participants focused on a fixation cue and were not explicitly instructed to rest.

## 2.2 RESTING STATE CONNECTIVITY ANALYSIS

The resting state analysis for both datasets was common. The preprocessing phase for the analysis involved using MNE-Python Gramfort et al. (2013) to process resting-state EEG data from each epoch spanning a time window from 0 to 2 seconds relative to the trial start onset. A continuous wavelet transform (CWT) using Morlet wavelets Tallon-Baudry et al. (1997) was then applied to decompose the EEG signals into these desired frequency bands: theta (4–8 Hz), alpha (8–13 Hz), and beta (13–30 Hz). To analyze steady-state connectivity patterns from resting-state EEG data, we employed spectral connectivity measures, including coherence (COH) and phase-locking value (PLV) Lachaux et al. (1999).

Spectral connectivity was estimated using Coherence (COH) and phase-locking value (PLV) as connectivity metrics to evaluate both amplitude and phase coupling between different brain regions.

Coherence measures the linear relationship between two signals in the frequency domain, capturing both the amplitude and phase coupling across frequency bands.

$$COH(f) = \frac{|E[S_{xy}(f)]|}{\sqrt{E[S_{xx}(f)] \cdot E[S_{yy}(f)]}} \tag{1}$$

The cross-spectrum $S_{xy}(f)$ is a measure of the spectral density of the correlation between two signals $x(t)$ and $y(t)$ at a specific frequency $f$. The auto-spectra $S_{xx}(f)$ and $S_{yy}(f)$ are the Fourier transforms of the autocorrelation functions of $x(t)$ and $y(t)$, respectively, and represent the power spectral densities of the signals.

Similarly, Phase-Locking Value (PLV) measures the consistency of the phase difference between two signals across multiple trials, independent of their amplitude. PLV ranges from 0 to 1, where 0 indicates no phase locking (random phase differences) and 1 indicates perfect phase synchronization (constant phase difference).

$$PLV = \left| E\left[ \frac{S_{xy}(f)}{|S_{xy}(f)|} \right] \right| \tag{2}$$

The resting state EEG analysis described above was performed on the segmented two-second-long time window from EEG data, and the resulting connectivity matrices were averaged across each participant trial to obtain a representation of functional connectivity.

## 2.3 MODEL ARCHITECTURE AND TRAINING

This study proposes a novel architecture that combines the feature extraction capabilities of EEGNet (Lawhern et al., 2018) with the adaptability of HyperNetworks (Ha et al., 2017) for motor imagery classification. This method uses a hypernetwork to generate adaptive weights for EEGNet, leveraging user-specific information from the resting state EEG data for cross-session and cross-user generalisation. Figure 1 depicts the model architecture and the learning mechanism.

### 2.3.1 EEGNET

EEGNet is a specialized neural network architecture designed to handle the unique characteristics of EEG signals. The model includes temporal and spatial convolutional layers optimized to capture relevant patterns from the multi-channel EEG data. Temporal convolutional layers focus on identifying patterns within the time domain of the signals, while spatial convolutional layers extract information based on the relationships between different EEG channels (Tshukahara, 2021). The EEGNet model was implemented using the Torcheeg framework (Zhang et al., 2024).

### 2.3.2 HYPERNETWORK

Hypernetworks are neural networks that generate the weights for another network (the main network: EEGNet) instead of learning them directly. For this study, the designed hypernetwork generated the weights of the core layers (conv2d and linear layers) of EEGNet. The hypernetwork (HyperNet) is a fully connected neural network with hidden layers of sizes 256 and 512, followed by a dropout with a probability of 0.3 to improve generalization. The resting state EEG data extracted from the two-second long time window are the inputs to this hypernetwork.

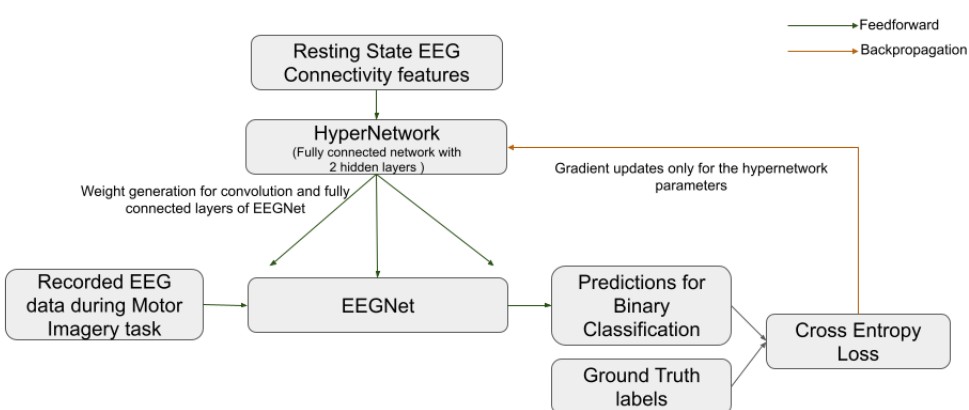

Figure 1: Overview of the proposed HyperEEGNet learning mechanism.

### 2.3.3 HYPEREEGNET TRAINING

The combined architecture, including the HyperNet and EEGNet, is trained as follows:

- HyperNet is used as a feedforward network to generate weights for EEGNet using resting-state connectivity data with dropout.

- Motor imagery activity data is extracted from a predefined time window (based on the experimental paradigm) in the raw data to perform the binary class classification with a forward pass on EEGNet with the generated weights from HyperNet.

- Cross entropy loss is accumulated for a batch of 50 epochs, and backpropagation is performed only on HyperNet parameters. Adam optimiser with learning rate 1e-4 is used.

### 2.4 EXPERIMENTAL SETUP AND PERFORMANCE EVALUATION

The experiment is set to evaluate two conditions: cross-session and cross-user for each dataset: BCI IV IIa and Dreyer et al. (2023) with a baseline comparison with EEGNet. The experiments were performed using the dataset from MOABB library (Aristimunha et al., 2023), and models were trained and evaluated using Torch and sci-kit-learn libraries.

### 2.4.1 CROSS-SESSION CONDITION

For the dataset from Dreyer et al. (2023), the "acquisition runs" from 33 participants are used for training and stratified 5-fold cross-validation is used to select the best model. Performance evaluation with accuracy metrics is performed for the "online" runs to evaluate HyperEEGNet compared to EEGNet.

For the BCI IV IIa dataset, the data from all nine participants is divided into five folds with stratified cross-validation; each fold in the iteration is considered as a test set while the other set is split with an 80-20 ratio to choose the best-performing model on the validation set. Accuracy metrics on the test set are evaluated for HyperEEGNet and compared with EEGNet.

### 2.4.2 CROSS-USER CONDITION

For the Leave-N-out (with N=8,16 and 32) strategy to test the HyperEEGNet performance compared to EEGNet, the "acquisition runs" from randomly selected (42-N) participants were used for training. 20% split is used as a validation set to select the best model. Performance evaluation with accuracy metrics is performed using data from the N participants for the "online" runs to evaluate HyperEEGNet compared to EEGNet. Analysis of such 100 random combinations reports the mean accuracy and standard deviation in Table 4 in the Appendix section. Non-parametric statistical tests

(Wilcoxon Signed Rank Test) recorded a statistically significant increase (p<0.005 for all N) in the performance using HyperEEGNet compared to EEGNet.

# 3 RESULTS

The results of the experiments, as presented in Table 1, report the performance of the proposed HyperEEGNet architecture compared to the baseline EEGNet in cross-session conditions on the Dreyer et al. (2023) dataset. For the cross-session condition, the HyperEEGNet again outperformed EEGNet, with a mean accuracy of 83.51% ± 0.68 compared to EEGNet's 75.87% ± 6.62.

| Iteration | Cross-Session Condition | |
|---|---|---|
| | HyperNet + EEGNet (%) | EEGNet (%) |
| 1 | 84.25 | 81.49 |
| 2 | 83.47 | 73.79 |
| 3 | 83.78 | 65.46 |
| 4 | 83.66 | 81.26 |
| 5 | 83.51 | 77.35 |
| Mean ± SD | 83.51 ± 0.68 | 75.87 ± 6.62 |

Table 1: Mean accuracy with standard deviation (SD) across five iterations of cross-session (on online runs) conditions on Dreyer et al. (2023) Dataset .

| Participant ID | HyperNet + EEGNet (%) | EEGNet (%) |
|---|---|---|
| 1 | 60.07 | 83.33 |
| 2 | 53.47 | 62.85 |
| 3 | 63.89 | 82.29 |
| 4 | 76.04 | 58.68 |
| 5 | 59.03 | 54.17 |
| 6 | 68.40 | 72.22 |
| 7 | 68.40 | 63.54 |
| 8 | 75.00 | 88.19 |
| 9 | 64.58 | 70.83 |
| Mean ± SD | 65.43 ± 07.40 | 70.68 ± 11.90 |

Table 2: Mean accuracy with standard deviation (SD) across five iterations of cross-user condition on BCI Competition IV IIa Dataset with Leave One Subject Out (LOSO) strategy.

| Iteration | HyperNet + EEGNet (%) | EEGNet (%) |
|---|---|---|
| 1 | 79.61 | 79.38 |
| 2 | 82.21 | 81.70 |
| 3 | 80.69 | 80.12 |
| 4 | 79.61 | 81.27 |
| 5 | 79.18 | 80.31 |
| Mean ± SD | 80.26 ± 1.23 | 80.56 ± 00.93 |

Table 3: Mean accuracy with standard deviation (SD) across five iterations of cross-session (using all 9 participants' data) condition for BCI Competition IV IIa Dataset.

The HyperEEGNet and baseline EEGNet models were evaluated using the Leave-One-Subject-Out (LOSO) strategy on the BCI Competition IV IIa dataset. The results represented in Table 2 indicate that while EEGNet achieved a higher overall mean accuracy (70.68% ± 11.90) compared to HyperEEGNet (65.43% ± 07.40), there were notable differences in performance for certain participants. For instance, HyperEEGNet outperformed EEGNet for Participant IDs 4, 5, and 7, with improvements of 17.36%, 4.86%, and 4.86%, respectively. However, for participants with higher

baseline performance (e.g., Participant IDs 1, 3, and 8), EEGNet achieved superior results. For the cross-session evaluation, the performance of HyperEEGNet and EEGNet was more comparable, as observed in Table 3, with mean accuracies of 80.26% ± 1.23 for HyperEEGNet and 80.56% ± 00.93 for EEGNet.

## 4 DISCUSSION

### 4.1 LEARNING REPRESENTATIONS FROM RESTING STATE

Experimental results indicate the unique possibility of leveraging resting state EEG data for learning downstream tasks like motor imagery classification. The comparison across two different sizes of datasets also confirms the positive outcomes of the efforts in the field to build large, robust datasets for training foundational models (Ferrante et al., 2024; Chen et al., 2024). During the training phase for HyperEEGNet architecture on the Dreyer et al. 2023 dataset, we also observed a steep learning curve, indicating a rapid convergence in ~50 epochs. The HyperNet was also prone to overfitting with a larger epoch size (500+) for training, especially in the case of cross-user conditions, where it was more evident. Notably, the smaller standard deviation for HyperEEGNet across the performance benchmarks indicates more stable performance across subjects than EEGNet.

Though the proposed approach focused on learning the adaptive weights for two class motor imagery classification, it opens up a future direction to generalise the learning mechanism across different downstream tasks or a larger number of classes for motor imagery.

### 4.2 INTERPRETING HYPEREEGNET

Apart from the performance metrics, some takeaways indicate potential as well as raise interesting questions in learning via the proposed approach. There are participants from both datasets known to have lower BCI performance across studies using different classifiers, and the contrary is true where few participants have consistently higher accuracies when trained specifically on the participant's data. For example, Participant ID 4 in the BCI Competition dataset has low BCI performance. However, Participant ID 3 has consistently high accuracies when using different classifiers. Surprisingly, the proposed approach performs well on Participant IDs 4, 5, and 7 but doesn't do well enough for Participant ID 3 in cross-user scenarios. These questions are open for exploration since they need to interpret the weights generated by the HyperNet; how do they compare with an EEGNet trained directly on activity data? Moreover, the resting state data can be represented in many ways; the proposed work did not explore optimising the representations for resting-state brain connectivity. There could be other important features useful for downstream tasks that are not captured in the connectivity measures.

### 4.3 TRANSFER LEARNING AND FEW-SHOT LEARNING

While the current approach can be considered an approach towards meta-learning by learning to learn weights of the downstream task, the work has not explored the paradigm of few-shot learning for faster adaptation compared to other existing approaches or the efficacy of this architecture compared with other transfer learning approaches. A benchmark against approaches for transfer learning and few-shot learning successful on EEG datasets is necessary to justify the approach holistically.

### 4.4 HYPERNETS FOR SMALLER FOOTPRINTS

Current work focused on successfully learning representations from resting state EEG data for motor imagery without optimising the size of the HyperNet. However, hypernetwork architectures are helpful for model compression. Efforts towards model compression without an impact on performance can be fruitful for real-world deployment of the BCI models. Task-specific information like restricting the input frequency bands and identifying efficient connectivity metrics can be an interesting future direction.

## 5 CONCLUSION

This work propose a novel HyperEEGNet architecture and introduces a promising new direction in EEG-based brain-computer interfaces (BCIs) by leveraging HyperNetworks to adaptively generate weights for EEGNet, utilizing resting-state EEG data for downstream motor imagery classification. This approach underscores the untapped potential of resting-state EEG, not only as a passive baseline to evaluate or correlate the BCI performance or illiteracy but as a source of user-specific features that enhance generalization across subjects and sessions. The positive results across cross-user and cross-session conditions indicate that resting-state data can be effectively harnessed for learning personalized representations in BCIs.

With focused efforts, instead of relying solely on task-related data, using resting-state data for model training can reduce the need for large amounts of labelled task data, which is often a bottleneck in real-world BCI applications. Furthermore, the architecture's rapid convergence and susceptibility to overfitting emphasize the need for further research into regularization techniques and adaptive training strategies specific to hypernetwork-based models. Looking forward, the findings suggest several key avenues for future exploration. This work sets the stage for more scalable, adaptive, and personalized BCIs, bridging the gap between laboratory research and practical, everyday use.

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

# A  APPENDIX

We plan to make our code publicly available on acceptance to ensure reproducibility and facilitate further research.

## A.1  LEAVE-N-OUT ANALYSIS FOR DREYER ET AL. 2023 DATASET

For the Leave-N-out (with N=8,16 and 32) strategy to test the HyperEEGNet performance compared to EEGNet, the "acquisition runs" from randomly selected (42-N) participants were used for training. 20% split is used as a validation set to select the best model. Performance evaluation with accuracy metrics is performed using data from the N participants for the "online" runs to evaluate HyperEEGNet compared to EEGNet. Analysis of such 100 random combinations reports the mean accuracy and standard deviation in Table 4. Non-parametric statistical tests (Wilcoxon Signed Rank Test) recorded a statistically significant increase ($p<0.005$ for all N) in the performance using HyperEEGNet compared to EEGNet.

| Number of participants in test set (N) | HyperNet + EEGNet (%) | EEGNet (%) |
|:---:|:---|:---|
| 8 | $84.10 \pm 02.11$ | $83.87 \pm 02.10$ |
| 16 | $84.86 \pm 01.02$ | $83.94 \pm 00.97$ |
| 32 | $76.47 \pm 02.00$ | $73.45 \pm 02.61$ |

Table 4: Mean accuracy with standard deviation (SD) across 100 combinations of cross-user condition on Dreyer et al. 2023 dataset with Leave N Subject Out strategy.

## A.2  SESSION-WISE ANALYSIS FOR BCI IV IIA DATASET

For the BCI IV IIa dataset, since there are just 9 participants, we use the Leave one subject out (LOSO) a strategy where, across nine folds, each participant's performance is evaluated while training on the first session data from 8 participants. Analysis reports the mean accuracy and standard deviation in Table 5 Accuracy metrics are evaluated on the second session's data of each participant left out during training. Non-parametric statistical tests (Wilcoxon Signed Rank Test) recorded a statistically significant decrease ($p<0.05$ for all N) in the performance using HyperEEGNet compared to EEGNet.

| Participant ID | HyperNet + EEGNet (%) | EEGNet (%) |
|---|---|---|
| 1 | 60.42 | 75.00 |
| 2 | 50.69 | 59.72 |
| 3 | 59.03 | 90.97 |
| 4 | 59.72 | 62.50 |
| 5 | 57.63 | 59.72 |
| 6 | 63.88 | 68.05 |
| 7 | 56.94 | 51.38 |
| 8 | 74.30 | 95.83 |
| 9 | 59.02 | 70.13 |
| Mean ± SD | 60.19 ± 06.35 | 70.37 ± 14.78 |

Table 5: Mean accuracy with standard deviation (SD) using test session for cross-user condition on BCI Competition IV IIa Dataset with Leave One Subject Out (LOSO) strategy.

### A.3 SESSION-WISE ANALYSIS FOR BCI IV IIB DATASET

BCI IV IIb dataset is different from BCI IV IIa since the number of EEG channels used for data recording are 3 compared to 22. Since there are just 9 participants, we use the Leave one subject out (LOSO) a strategy where, across nine folds, each participant's performance is evaluated while training on the first session data from 8 participants. Analysis reports the mean accuracy and standard deviation in Table 6 Accuracy metrics are evaluated on the second session's data of each participant left out during training. Non-parametric statistical tests (Wilcoxon Signed Rank Test) recorded a statistically significant decrease ($p < 0.05$ for all N) in the performance using HyperEEGNet compared to EEGNet.

| Participant ID | HyperNet + EEGNet (%) | EEGNet (%) |
|---|---|---|
| 1 | 63.12 | 75.31 |
| 2 | 49.64 | 57.50 |
| 3 | 51.87 | 54.06 |
| 4 | 83.12 | 86.25 |
| 5 | 57.50 | 78.12 |
| 6 | 54.06 | 79.06 |
| 7 | 56.87 | 72.18 |
| 8 | 85.00 | 79.37 |
| 9 | 60.62 | 88.75 |
| Mean ± SD | 62.42 ± 12.94 | 74.51 ± 11.78 |

Table 6: Mean accuracy with standard deviation (SD) using test session for cross-user condition on BCI Competition IV IIb Dataset with Leave One Subject Out (LOSO) strategy.

