# OpenReview forum: "From Rest to Action: Adaptive Weight Generation for Motor Imagery Classification from Resting-State EEG Using Hypernetworks"
_ICLR.cc/2025/Conference — Submitted to ICLR 2025_

### Official Review · Reviewer_qyVC · 2024-10-29

**Soundness:** 2
**Presentation:** 1
**Contribution:** 2
**Rating:** 3
**Confidence:** 5

**Summary:**

The authors propose a novel architecture, HyperEEGNet, to improve EEG-based brain-computer interface (BCI) systems, addressing the limitations of long calibration sessions and the underutilization of resting-state EEG data in motor imagery decoding tasks. By integrating HyperNetworks with the EEGNet architecture, HyperEEGNet adaptively generates weights for motor imagery classification based on resting-state data. In Leave-Subject-Out scenarios using a dataset with nine participants, the model performs comparably to the baseline EEGNet. However, when scaled to datasets with 33 participants, HyperEEGNet demonstrates enhanced generalization capabilities, effectively leveraging resting-state EEG information to handle unseen subjects. The model achieves robust representations in both cross-session and cross-user scenarios, highlighting the potential of resting-state data for downstream tasks like motor imagery classification. Furthermore, the findings indicate that HyperEEGNet's smaller footprint reduces memory and storage requirements, making it suitable for edge computing. This approach promises faster user calibration and improved feasibility for real-world BCI applications, advancing the field significantly.

**Strengths:**

Robust Generalization: The model demonstrates strong generalization capabilities, performing well in both Leave-Subject-Out scenarios and with larger datasets, indicating its effectiveness in handling unseen subjects.

Reduced Calibration Time: The approach promises faster user calibration, which is crucial for real-world BCI applications, making it more user-friendly.

**Weaknesses:**

Limited Dataset Size: The initial evaluations involve a relatively small dataset with only nine participants, which may not fully capture the variability present in broader populations. The generalizability of the findings could be questioned without larger, more diverse datasets.

Dependence on Resting-State Data: While leveraging resting-state EEG data is innovative, the model's effectiveness might be limited if the quality or relevance of the resting-state data varies across users or sessions.

Complexity of HyperNetworks: The integration of HyperNetworks may introduce additional complexity in model training and tuning, potentially requiring more computational resources and expertise to implement effectively.

Interpretability: As with many deep learning models, the interpretability of HyperEEGNet's decision-making process might be limited, making it challenging to understand how specific features influence classifications.

**Questions:**

What specific strategies were implemented to mitigate overfitting during training, especially given the observed risks at larger epoch sizes? How do you plan to validate the model's performance in cross-user scenarios beyond the training dataset?

Could you elaborate on the implications of a steep learning curve and rapid convergence in just 50 epochs? What does this suggest about the model's capacity to capture complex patterns in the data?

While the focus on two-class motor imagery classification is noted, what are the plans for extending this model to accommodate multiple classes or different downstream tasks? How do you envision addressing potential challenges in this expansion?

 How do you explain the performance variations among participants, particularly the discrepancy in accuracy for Participant ID 3? What insights can be gained from comparing the weights generated by the HyperNet with those from an EEGNet trained directly on activity data?

Why was the optimization of resting-state EEG data representations not explored, particularly regarding brain connectivity? What additional features do you think might be important for downstream tasks that are not captured by current measures?

What criteria will you use to compare the efficacy of HyperEEGNet against other transfer learning approaches? Are there specific metrics or datasets that you consider critical for this comparison?

What specific task-related information do you believe should be incorporated to optimize the input frequency and further enhance model performance? How will this impact the model's practical deployment?

---

> ### Author Response · Authors · 2024-11-28
>
> We thank the reviewer for sharing their suggestions and appreciate their acknowledgement of the strengths of our work. Based on the reviews, we have updated the Appendix Section in the original submission, and the updated document can be viewed in the submission.
>
> Following are the responses to the reviewer’s comments:
>
> _Limited Dataset Size: The initial evaluations involve a relatively small dataset with only nine participants, which may not fully capture the variability present in broader populations. The generalizability of the findings could be questioned without larger, more diverse datasets._
>
> We provide evaluation using two standard EEG datasets used in the benchmarks and evaluations: BCI IV IIa with 9 participants and Dreyer et al. with 42 participants. As reviewers suggested, we have also added three more evaluations to validate our approach. The results are summarised and included in the Appendix section.
>
> _Dependence on Resting-State Data: While leveraging resting-state EEG data is innovative, the model's effectiveness might be limited if the quality or relevance of the resting-state data varies across users or sessions._
>
> The core idea of our work is to include resting state data and extract consistent and unique features for each participant. This approach is motivated by the previous findings showing a correlation between resting state markers and BCI performance. The model can adapt to the across-user variability and generalise using such features.
> We agree with the concern of quality across sessions for the same users. Citing this in our original submission, we have included resting state data preceding each trial to accommodate and build a robust model for such variations in the data.
>
> _Complexity of HyperNetworks: The integration of HyperNetworks may introduce additional complexity in model training and tuning, potentially requiring more computational resources and expertise to implement effectively._
>
> We agree with the reviewer's concern about complexity and implementation; however, the generalisability of BCIs is more rewarding in terms of practical out-of-the-lab applications in our understanding against the cost of longer training time and minor increment in inference time. Moreover, using optimised hypernetworks, memory footprint can be reduced considerably since they reduce the storage requirements for the model weights.
>
> _Interpretability: As with many deep learning models, the interpretability of HyperEEGNet's decision-making process might be limited, making it challenging to understand how specific features influence classifications._
>
> We agree with the reviewer’s concern and interpretability and raised our concern in the original submission: Section 4.2, Page number 6 295-308. While the interpretation of weights generated using Hypernetwork is not compared, EEGNet and the use of resting-state data have been thoroughly interpreted, and their neurophysiological basis is validated. We list a few references that are cited in the original submission that support the work:
>
> This work interprets the weights learnt by EEGNet across different paradigms using EEG and validates against the known markers in neurophysiology.
>
> Vernon J Lawhern, Amelia J Solon, Nicholas R Waytowich, Stephen M Gordon, Chou P Hung, and
> Brent J Lance. Eegnet: a compact convolutional neural network for eeg-based brain–computer
> interfaces. Journal of neural engineering, 15(5):056013, 2018.
>
> Following works have validated the correlation between EEG resting state markers and BCI performance on motor imagery:
>
> Eidan Tzdaka, Camille Benaroch, Camille Jeunet, and Fabien Lotte. Assessing the relevance of
> neurophysiological patterns to predict motor imagery-based bci users’ performance. In 2020
> IEEE International Conference on Systems, Man, and Cybernetics (SMC), pp. 2490–2495, 2020. doi: 10.1109/SMC42975.2020.9283307.
>
> David Trocellier, Bernard N’Kaoua, and Fabien Lotte. Validating neurophysiological predictors of
> bci performance on a large open source dataset. In 9th Graz Brain-Computer Interface Conference 2024-GBCIC2024, 2024.
>
> Benjamin Blankertz, Claudia Sannelli, Sebastian Halder, Eva M Hammer, Andrea K¨ubler, Klaus-Robert M¨uller, Gabriel Curio, and Thorsten Dickhaus. Neurophysiological predictor of smr-based bci performance. Neuroimage, 51(4):1303–1309, 2010.

---

> > ### Author Response · Authors · 2024-11-28
> >
> > _What specific strategies were implemented to mitigate overfitting during training, especially given the observed risks at larger epoch sizes? How do you plan to validate the model's performance in cross-user scenarios beyond the training dataset?_
> >
> > Dropout and Weight decay were implemented to mitigate overfitting. Since the best model was chosen based on validation accuracy, the results on the test set were robust against overfitting even if the epoch sizes were larger.
> > Cross-dataset evaluation can be used to validate performance beyond the current dataset used for training. However, the major challenge is the varying number of channels. Choosing a subset of identical channels across datasets and evaluating the performance could be a valuable experiment for the future.
> >
> > _Could you elaborate on the implications of a steep learning curve and rapid convergence in just 50 epochs? What does this suggest about the model's capacity to capture complex patterns in the data?_
> >
> > We observed that the HyperEEGNet converged with fewer epochs (<200), while EEGNet required 400-500 epochs on the Dreyer et al. dataset. While the observation is interesting, to comment on the implications or understand the learning mechanisms, the next step is to interpret and compare the EEGNet weights/activations generated using Hypernetworks with resting-state data against EEGNet with only activity data.
> >
> > _While the focus on two-class motor imagery classification is noted, what are the plans for extending this model to accommodate multiple classes or different downstream tasks? How do you envision addressing potential challenges in this expansion?_
> >
> > After updated analysis in the Appendix section, our current work includes 3 standard datasets in the motor imagery domain that include the variability in the number of channels, number of participants and available hardware. Datasets with more participants and activity classes will benefit such evaluation. Moreover, the approach can have EEG datasets from other paradigms, like inner speech recognition, that are prone to cross-subject variability. However, validating the interpretability is a potential challenge while expanding to other paradigms. Addressing the neurophysiological interpretation of generated weights and activations using HyperEEGNet is an essential next step.
> >
> > _How do you explain the performance variations among participants, particularly the discrepancy in accuracy for Participant ID 3? What insights can be gained from comparing the weights generated by the HyperNet with those from an EEGNet trained directly on activity data?_
> >
> > Referring to the BCI IV IIa dataset, we observed that EEGNet consistently performed better than HyperEEGNet. While the observation is contradictory, to comment on the implications or understand the learning mechanisms, the next step is to interpret and compare the EEGNet weights/activations generated using Hypernetworks with resting-state data against EEGNet with only activity data. We also tried using data from the 2nd session, where the performance of HyperEEGNet was not at par with EEGNet.
> >
> >
> > _Why was the optimization of resting-state EEG data representations not explored, particularly regarding brain connectivity? What additional features do you think might be important for downstream tasks that are not captured by current measures?_
> >
> > We assumed that the preliminary analysis and the feasibility can be validated using whole-brain connectivity based on the number of channels from a particular dataset. We also used all frequency bands relevant to the motor imagery paradigm (mu and beta bands 8-32 Hz). However, one of the previous works has cited the use of gamma band as the predictor of the BCI performance however the correlation was not as strong.
> > We understand that the current work can validate the hypernetwork architecture by learning user-specific representations. Future work can explore optimisations and model interpretability by focusing on the neurophysiological perspectives.

---

> > > ### Author Response · Authors · 2024-11-28
> > >
> > > _What criteria will you use to compare the efficacy of HyperEEGNet against other transfer learning approaches? Are there specific metrics or datasets that you consider critical for this comparison?_
> > >
> > > We thank the reviewers for highlighting the necessity of standardising such benchmarks in EEG classification tasks. We understand that Leave-one-subject-out and Leave-N-subject-out strategies are the best evaluation techniques across datasets. We follow those strategies to evaluate and compare the current performances. Most of the proposed techniques in transfer learning using data alignment can be combined with the current approach to validate the efficacy.
> > >
> > > Most of the work in transfer learning focuses, on a few shot analyses. We discuss that perspective in section 4.3 Page 6, 310-315 and consider the limitations of the current work that can be explored in the future.
> > > Based on the reviews, we have also listed the current state of the art using Leave-One-Subject-Out for multiple datasets. We also evaluate the Dreyer et al. dataset using Leave-N-Subjects-Out to make our experiments more robust when choosing subjects for the test set. We understand that our current approach validates the most challenging aspect of transfer learning, i.e. zero shot analysis.
> > >
> > > Our current work after updated analysis in the Appendix section includes 3 standard datasets in the motor imagery domain that includes the variability in number of channels, number of participants and available hardware. More datasets with a larger number of participants will be beneficial for such evaluation. Moreover, the approach can have EEG datasets from other paradigms, like inner speech recognition, that are prone to cross-subject variability.
> > >
> > > _What specific task-related information do you believe should be incorporated to optimize the input frequency and further enhance model performance? How will this impact the model's practical deployment?_
> > >
> > > We understand that the reviewer has referred to using task-related information from resting state EEG for hypernetwork input to optimise performance further. However, we request the reviewer to clarify the context of the term input frequency.
> > >
> > > Using source-level information, i.e., connectivity across specific brain regions, instead of using sensor-level details, can be more useful and interpretable.
> > > Moreover, previous works have identified a correlation with band power in the gamma band (55-85 Hz), which this work has not explored. We understand that choosing specific features from the resting state data doesn't impact the intended practical deployment as far as the paradigm includes recording resting state data.

---

> ### Author Response · Authors · 2024-12-02
>
> Dear Reviewer qyVC
>
> We kindly follow up on your feedback on our manuscript since today is the last discussion day.
>
> In our earlier responses, we believe we have addressed your concerns comprehensively. We are eager to know if there are any additional suggestions or specific points we could consider to enhance our manuscript further.
>
> We sincerely hope you might provide us with further insights that could guide us in strengthening our work.
>
> Thank you for your time and consideration.

---

### Official Review · Reviewer_Fidh · 2024-10-31

**Soundness:** 2
**Presentation:** 1
**Contribution:** 2
**Rating:** 3
**Confidence:** 5

**Summary:**

The authors introduce a new architecture called HyperEEGNet aimed at enhancing EEG-based brain-computer interface (BCI) systems. This innovation addresses issues related to lengthy calibration sessions and the limited use of resting-state EEG data in motor imagery decoding tasks. By combining HyperNetworks with the EEGNet framework, HyperEEGNet adaptively generates weights for motor imagery classification based on resting-state data. In Leave-Subject-Out scenarios using a dataset of nine participants, its performance is comparable to the baseline EEGNet. However, when applied to larger datasets with 33 participants, HyperEEGNet shows improved generalization capabilities, effectively utilizing resting-state EEG information to manage unseen subjects. The model provides strong representations in both cross-session and cross-user contexts, underscoring the value of resting-state data for tasks such as motor imagery classification. Additionally, the results indicate that HyperEEGNet has a smaller memory and storage footprint, making it well-suited for edge computing. This approach offers faster user calibration and enhances the practicality of real-world BCI applications, representing a significant advancement in the field.

**Strengths:**

The model exhibits robust generalization capabilities, performing effectively in both Leave-Subject-Out scenarios and with larger datasets, demonstrating its ability to handle unseen subjects. Additionally, this approach promises reduced calibration time, which is essential for real-world BCI applications, thereby enhancing user-friendliness and practicality.

**Weaknesses:**

The initial evaluations rely on a relatively small dataset comprising just nine participants, which may not adequately reflect the variability found in larger populations. This raises questions about the generalizability of the findings without access to more extensive and diverse datasets. Additionally, while the use of resting-state EEG data is a novel approach, the model's performance may be affected if the quality or relevance of this data varies among different users or sessions. Furthermore, incorporating HyperNetworks adds a layer of complexity to the training and tuning process, potentially necessitating greater computational resources and specialized knowledge for effective implementation. Lastly, like many deep learning models, HyperEEGNet may have limitations in interpretability, making it difficult to ascertain how specific features impact its classification decisions.

**Questions:**

What specific task-related information do you think should be included to optimize the input frequency and enhance model performance? How might this affect the model's practical deployment?

Why wasn't the optimization of resting-state EEG data representations, especially concerning brain connectivity, explored? What additional features do you believe are important for downstream tasks that current measures do not capture?

What criteria will you use to evaluate the efficacy of HyperEEGNet in comparison to other transfer learning methods? Are there particular metrics or datasets that you consider essential for this assessment?

---

> ### Author Response · Authors · 2024-11-28
>
> We thank the reviewer for sharing their suggestions and appreciate their acknowledgement of the strengths of our work. Based on the reviews, we have updated the Appendix Section in the original submission, and the updated document can be viewed in the submission.
>
> Following are the responses to the reviewer’s comments:
>
> _The initial evaluations rely on a relatively small dataset comprising just nine participants, which may not adequately reflect the variability found in larger populations. This raises questions about the generalizability of the findings without access to more extensive and diverse datasets._
>
> We provide evaluation using two standard EEG datasets used in the benchmarks and evaluations: BCI IV IIa with 9 participants and Dreyer et al. with 42 participants. As reviewers suggested, we have also added evaluation to validate our approach with leave N out to simulate different sizes of datasets. The results are summarised and included in the Appendix section of the revised submission.
>
>
> _Additionally, while using resting-state EEG data is a novel approach, the model's performance may be affected if the quality or relevance of this data varies among different users or sessions._
>
> The core idea of our work is to include resting state data and extract consistent and unique features for each participant. This approach is motivated by the previous findings showing a correlation between resting state markers and BCI performance. The model can adapt to the across-user variability and generalise using such features.
> We agree with the concern of quality across sessions for the same users. Citing this in our original submission, we have included resting state data preceding each trial to accommodate and build a robust model for such variations in the data.
>
> _Furthermore, incorporating HyperNetworks adds a layer of complexity to the training and tuning process, potentially necessitating greater computational resources and specialized knowledge for effective implementation._
>
> We agree with the reviewer's concern about complexity and implementation; however, the generalisability of BCIs is more rewarding in terms of practical out-of-the-lab applications in our understanding against the cost of longer training time and minor increment in inference time. Moreover, using optimised hypernetworks, memory footprint can be reduced considerably since they reduce the storage requirements for the model weights.
>
> _Lastly, like many deep learning models, HyperEEGNet may have limitations in interpretability, making it difficult to ascertain how specific features impact its classification decisions._
>
> We agree with the reviewer’s concern and interpretability and raised our concern in the original submission: Section 4.2, Page number 6 295-308. While the interpretation of weights generated using hypernets is not compared, EEGNet and the use of resting-state data have been thoroughly interpreted, and their neurophysiological basis is validated. We list a few references that are cited in the original submission that support the work:
>
> This work interprets the weights learnt by EEGNet across different paradigms using EEG and validates against the known markers in neurophysiology.
>
> Vernon J Lawhern, Amelia J Solon, Nicholas R Waytowich, Stephen M Gordon, Chou P Hung, and
> Brent J Lance. Eegnet: a compact convolutional neural network for eeg-based brain–computer
> interfaces. Journal of neural engineering, 15(5):056013, 2018.
>
> Following works have validated the correlation between EEG resting state markers and BCI performance on motor imagery:
>
> Eidan Tzdaka, Camille Benaroch, Camille Jeunet, and Fabien Lotte. Assessing the relevance of
> neurophysiological patterns to predict motor imagery-based bci users’ performance. In 2020
> IEEE International Conference on Systems, Man, and Cybernetics (SMC), pp. 2490–2495, 2020. doi: 10.1109/SMC42975.2020.9283307.
>
> David Trocellier, Bernard N’Kaoua, and Fabien Lotte. Validating neurophysiological predictors of
> bci performance on a large open source dataset. In 9th Graz Brain-Computer Interface Conference 2024-GBCIC2024, 2024.
>
> Benjamin Blankertz, Claudia Sannelli, Sebastian Halder, Eva M Hammer, Andrea K¨ubler, Klaus-Robert M¨uller, Gabriel Curio, and Thorsten Dickhaus. Neurophysiological predictor of smr-based bci performance. Neuroimage, 51(4):1303–1309, 2010.

---

> > ### Author Response · Authors · 2024-11-28
> >
> > _What specific task-related information do you think should be included to optimize the input frequency and enhance model performance? How might this affect the model's practical deployment?_
> >
> > We understand that the reviewer has referred to using task-related information from resting state EEG for hypernetwork input to optimise performance further. However, we request the reviewer to clarify the context of the term input frequency.
> >
> > Using source-level information, i.e., connectivity across specific brain regions, instead of using sensor-level details, can be more useful and interpretable.
> > Moreover, previous works have identified a correlation with band power in the gamma band (55-85 Hz), which this work has not explored. We understand that choosing specific features from the resting state data doesn't impact the intended practical deployment as far as the paradigm includes recording resting state data.
> >
> > _Why wasn't the optimization of resting-state EEG data representations, especially concerning brain connectivity, explored? What additional features do you believe are important for downstream tasks that current measures do not capture?_
> >
> > We assumed that the preliminary analysis and the feasibility can be validated using whole-brain connectivity based on the number of channels from a particular dataset. We also used all frequency bands relevant to the motor imagery paradigm (mu and beta bands 8-32 Hz). However, one of the previous works has cited the use of gamma band as the predictor of the BCI performance however the correlation was not as strong.
> > We understand that the current work can validate the hypernetwork architecture by learning user-specific representations. Future work can explore optimisations and model interpretability by focusing on the neurophysiological perspectives.
> >
> >
> > _What criteria will you use to evaluate the efficacy of HyperEEGNet in comparison to other transfer learning methods? Are there particular metrics or datasets that you consider essential for this assessment?_
> >
> > We thank the reviewers for highlighting the necessity of standardising such benchmarks in EEG classification tasks. We understand that Leave-one-subject-out and Leave-N-subject-out strategies are the best evaluation techniques across datasets. We follow those strategies to evaluate and compare the current performances. Most of the proposed techniques in transfer learning using data alignment can be combined with the current approach to validate efficacy.
> >
> > Most of the work in transfer learning focuses, on a few shot analyses. We discuss that perspective in section 4.3 Page 6, 310-315 and consider the limitations of the current work that can be explored in the future.
> > Based on the reviews, we have also listed the current state of the art using Leave-One-Subject-Out for multiple datasets. We also evaluate the Dreyer et al. dataset using Leave-N-Subjects-Out to make our experiments more robust when choosing subjects for the test set. We understand that our current approach validates the most challenging aspect of transfer learning, i.e. zero shot analysis.
> >
> > Our current work, after updated analysis in the Appendix section, includes 3 standard datasets in the motor imagery domain that include the variability in the number of channels, number of participants and available hardware. More datasets with a larger number of participants will be beneficial for such evaluation. Moreover, the approach can have EEG datasets from other paradigms, like inner speech recognition, that are prone to cross-subject variability.

---

> ### Author Response · Authors · 2024-12-02
>
> Dear Reviewer Fidh
>
> We kindly follow up on your feedback on our manuscript since today is the last discussion day.
>
> In our earlier responses, we believe we have addressed your concerns comprehensively. We are eager to know if there are any additional suggestions or specific points we could consider to enhance our manuscript further.
>
> We sincerely hope you might provide us with further insights that could guide us in strengthening our work.
>
> Thank you for your time and consideration.

---

### Official Review · Reviewer_CxNP · 2024-11-04

**Soundness:** 3
**Presentation:** 2
**Contribution:** 2
**Rating:** 5
**Confidence:** 4

**Summary:**

The paper aims to show the benefits of using a HyperNet architecture to improve the generalization capabilities of EEGNet for generalization given a large dataset. The authors also use data from the resting state before a trial as a novel input for motor imagery classification.

**Strengths:**

The paper is original in that they apply a HyperNet architecture to improve the generalization capabilities of an EEG motor imagery classifier. The paper evaluates inter-subject and inter-session performance, which are both important metrics for deployment of a BCI. The authors are fairly clear in how experiments are done, although I had some questions about intersession evaluation for the BCI IV IIa dataset. The work is significant in that a new method is evaluated on EEG data and shows strong generalization performance in a dataset with 42 subjects.

**Weaknesses:**

1. Claims on the strength of HyperNet + EEGNet would be improved through using a more comprehensive evaluation on the Dreyer et al. dataset. Leave-N-subjects-out train-test split should be done where around a quarter of the subjects are used as test subjects each time.
2. The HyperNet + EEGNet approach does not seem to work for the BCI IV IIa dataset (one of the two datasets tested). The authors mention that it is evidence that the method does not seem to work unless with a larger dataset. It would be better if there were another dataset that can be tested to show that the HyperNet + EEGNet approach does indeed improve classification given more than 9 subjects. Alternatively, the Dreyer et al. dataset could be evaluated while varying the number of subjects for training, e.g. 8, 16, 24, 32, etc to see if the trend of improving performance given more subjects occurs.

**Questions:**

1. The term "epoch" seems to be overloaded since they are common terms used in EEG and in machine learning but mean different things. I think it becomes unclear which meaning you use in the paper sometimes, for example here is the term "epoch" used in close proximity, although the former seems to mean a window of data and the latter seems to mean the number of times the HyperNet is trained on all batches:
>• Motor imagery activity data in the form of an epoch is used to perform the binary class
classification with a forward pass on EEGNet with the generated weights from HyperNet.
• Cross entropy loss is accumulated for a batch of 50 epochs, and backpropagation is performed only on HyperNet parameters. Adam optimiser with learning rate 1e-4 is used."

It would be better if the term "epoch" is better clarified when used.

2. >"For the dataset from Dreyer et al. (2023), the ”acquisition runs” from 33 participants are used for training and stratified 5-fold cross-validation is used to select the best model."

What variables are changed to select the best model? Is it the model architecture? Are hyperparameters tuned at all?

3. Although this passage is from the "2.4.1 Cross-Session Condition", it seems to imply that the train-test split is not split across sessions:
>"For the BCI IV IIa dataset, the data from all nine participants is divided into five folds with stratified cross-validation; each fold in the iteration is considered as a test set while the other set is split with an 80-20 ratio to choose the best-performing model on the validation set."

The original work for the BCI competition seems to imply that there are two sessions for each subject. Is there a reason that evaluation across sessions does not seem to be done in the current work?

4. In Table 1, EEGNet without the HyperNet seems to have 4 to 6 times the standard deviation as EEGNet with the HyperNet. Is there an explanation for this, especially when compared to how the ratio of the standard deviation seems to be much closer to 1 for Tables 2 and 3? If the same test in Table 1 is run with multiple seeds and using different subjects (not just the last 9 subjects) as the test set, would we still see such high variation across multiple folds for EEGNet without the HyperNet?

5. For a practical control interface during deployment, it seems unclear when the resting state would occur, which is the input used in the HyperNet. Would a separate resting state classifier have to be used? Or would some other heuristic to determine resting state be sufficient?

6. What is the current state of the art in terms of performance for these two datasets for the metrics you evaluated?

---

> ### Author Response · Authors · 2024-11-28
>
> We thank the reviewer for their insightful comments and suggestions. We appreciate their efforts in making this work more meaningful and robust. Based on the reviews, we have updated the Appendix Section in the original submission and the updated document can be viewed in the submission.
>
> We thank the reviewer for acknowledging the originality and novelty in using resting state data to test the generalisation capabilities of the hypernetwork architecture.
>
> Following are the responses to the reviewer’s comments:
>
> _Claims on the strength of HyperNet + EEGNet would be improved through using a more comprehensive evaluation on the Dreyer et al. dataset. Leave-N-subjects-out train-test split should be done where around a quarter of the subjects are used as test subjects each time._
>
> We perform the Leave-N-subject tests on Dreyer et al. dataset with N = 8,16,32. The results and the method are described in the Appendix section of the revised submission.
>
> _The HyperNet + EEGNet approach does not seem to work for the BCI IV IIa dataset (one of the two datasets tested). The authors mention that it is evidence that the method does not seem to work unless with a larger dataset. It would be better if there were another dataset that can be tested to show that the HyperNet + EEGNet approach does indeed improve classification given more than 9 subjects._
>
> We agree with the reviewers suggestion and evaluate the approach on BCI IV IIb dataset with 9 participants. The results are detailed in the Appendix section. EEGNet outperforms the HyperEEGNet significantly. We also consider that since this dataset consists of just 3 channels, resting state data might not reflect the whole brain connectivity of an individual. However, we leave the interpretation for future work, including the neurophysiological basis of model interpretability.
>
> _Alternatively, the Dreyer et al. dataset could be evaluated while varying the number of subjects for training, e.g. 8, 16, 24, 32, etc to see if the trend of improving performance given more subjects occurs._
>
> Agreeing with the reviewer's suggestion, we perform the leave-N-subject out where the number of training subjects changes as we leave more subjects out for the test set. The results and methodology are described in the Appendix section.
>
> _It would be better if the term "epoch" is better clarified when used._
>
> We thank the reviewer for pointing out the confusing terminology. We rewrite the statement as follows and also modify other instances across the document to avoid the confusion.
>
> Motor imagery activity data is extracted from a predefined time window (based on the experimental paradigm) in the raw data to perform the binary class classification with a forward pass on EEGNet with the generated weights from HyperNet. • Cross entropy loss is accumulated for a batch of 50 epochs, and backpropagation is performed only on HyperNet parameters. Adam optimiser with learning rate 1e-4 is used
>
> _"For the dataset from Dreyer et al. (2023), the ”acquisition runs” from 33 participants are used for training and stratified 5-fold cross-validation is used to select the best model."
> What variables are changed to select the best model? Is it the model architecture? Are hyperparameters tuned at all?_
>
> We tuned the architecture by changing the width of the two hidden layers of the hypernetwork. Hyperparameters like the learning rate for hypernet, number of epochs, and dropout probability were tuned based on the cross-validation performance.
>
> _Although this passage is from the "2.4.1 Cross-Session Condition", it seems to imply that the train-test split is not split across sessions:
> "For the BCI IV IIa dataset, the data from all nine participants is divided into five folds with stratified cross-validation; each fold in the iteration is considered as a test set while the other set is split with an 80-20 ratio to choose the best-performing model on the validation set."
> The original work for the BCI competition seems to imply that there are two sessions for each subject. Is there a reason that evaluation across sessions does not seem to be done in the current work?_
>
> There is no reason precisely not to follow the evaluation across sessions. We used a MOABB instance that loads both sessions by default as a dataset. We used them as a complete dataset to evaluate the results. However, we agree with the reviewer’s suggestion to standardise the results for comparison. The results for the across sessions are evaluated and reported in the Appendix.

---

> > ### Author Response · Authors · 2024-11-28
> >
> > _In Table 1, EEGNet without the HyperNet has 4 to 6 times the standard deviation as EEGNet with the HyperNet. Is there an explanation for this, especially when compared to how the ratio of the standard deviation seems to be much closer to 1 for Tables 2 and 3? If the same test in Table 1 is run with multiple seeds and using different subjects (not just the last 9 subjects) as the test set, would we still see such high variation across multiple folds for EEGNet without the HyperNet?_
> >
> > We agree with the reviewer’s concern and used different seeds to perform a similar analysis. The results showed a high deviation for a particular seed value (42). Therefore, we report the analysis with Leave N out analysis instead.
> >
> > _It seems unclear when the resting state would occur for a practical control interface during deployment, which is the input used in the HyperNet. Would a separate resting-state classifier have to be used? Or would some other heuristic to determine resting state be sufficient?_
> >
> > For practical purposes, control interfaces often include visual stimuli/feedback. BCIs may rely on time-locked activity guided by such visual stimuli, where the participant is asked to initiate a movement after a specific rest period.
> > On the other hand, BCIs can be event-locked i.e. when a particular event is triggered, the classification of motor imagery activity starts. A classifier can then detect the resting state before the initiation of motor imagery classification.
> > Moreover, an exciting direction would be to use resting-state data recorded once for each participant during the calibration phase. If the approach is successful during deployment, one may not need the resting state data every time before the motor imagery activity is performed.
> >
> >
> > _What is the current state of the art regarding the performance of these two datasets for the metrics you evaluated?_
> >
> > SOTA on Dreyer et al. using LOSO:
> >
> > *Wang et al. 2024*	(Use 85 participants from Dataset A and B)
> >
> > 75.25%
> >
> >
> > *Wimpff et al 2024* (Use 78 participants, using online mode with 1s time windows)
> >
> > 69.29 ± 13.70%
> >
> >
> > * Ouahidi et al 2024 *(Details on inclusion/exclusion or time windows are not clarified)
> >
> > 89.3 (Offline)
> >
> > 77.5 (Online)
> >
> >
> >
> >
> > Wang, Yihan, et al. "TFTL: A Task-Free Transfer Learning Strategy for EEG-based Cross-Subject & Cross-Dataset Motor Imagery BCI." IEEE Transactions on Biomedical Engineering (2024).
> >
> > Wimpff, Martin, et al. "Towards calibration-free online EEG motor imagery decoding using Deep Learning." ESANN, 2024.
> >
> > El Ouahidi, Yassine, et al. "Unsupervised Adaptive Deep Learning Method For BCI Motor Imagery Decoding." 2024 32nd European Signal Processing Conference (EUSIPCO). IEEE, 2024.
> >
> >
> > SOTA on BCI IV IIa using LOSO:
> >
> > *MI-DAGSC by Zhang el al. (2023)*					79.63 ± 12.27
> > (Train on session 1 and test on session 2)
> >
> > Zhang, Dongxue, et al. "MI-DAGSC: A domain adaptation approach incorporating comprehensive information from MI-EEG signals." Neural Networks 167 (2023): 183-198.
> >
> > All the methods evaluated on both datasets using the LOSO strategy use domain adaptation techniques by aligning data from the target subjects. For better performance on datasets, such approaches can be combined with a complex architecture and the hypernet approach proposed here.

---

> ### Author Response · Authors · 2024-12-02
>
> Dear Reviewer CxNP
>
> We kindly follow up on your feedback on our manuscript since today is the last discussion day.
>
> In our earlier responses, we believe we have addressed your concerns comprehensively. We are eager to know if there are any additional suggestions or specific points we could consider to enhance our manuscript further.
>
> We sincerely hope you might provide us with further insights that could guide us in strengthening our work.
>
> Thank you for your time and consideration.

---

> > ### Comment · Reviewer_CxNP · 2024-12-02
> >
> > Hello, I appreciate that the answers to the questions were clear and that additional evaluations were made to address some of the high variability in results. However, primarily due to the lack of more results that show HyperEEGNet performs better than EEGNet in more scenarios or cases, my score remains the same.

---

### Official Review · Reviewer_gs65 · 2024-11-04

**Soundness:** 1
**Presentation:** 1
**Contribution:** 1
**Rating:** 1
**Confidence:** 5

**Summary:**

In this paper, the authors proposed a HyperEEGNet architecture by combining the conventional HyperNetwork and EEGNet to adress cross-user variability for MI-BCI systems. The authors compared the performance of the proposed HyperEEGNet with that of competing EEGNet on various publicly available MI-EEG datasets in both cross-session and cross-user conditions.

**Strengths:**

This study try to address the important issue of cross-user variability and BCI illiteracy issues in the MI-EEG analysis by adopting the ability of HyperNetwork to adaptive weight generation to learn user-specific representations.

**Weaknesses:**

There is no substantial innovation in proposed method combining the conventional HyperNetworks and EEGNet.

The performance improvement of the proposed method over existing EEGNet has not been consistently demonstrated across multiple datasets. This is, the proposed model achieved improved performance on the Dreyer et al. dataset, while its performance degraded on the BCI Competition IV IIa dataset. Furthermore, there has been no meaningful discussion about these conflicting results.

No comparisons were conducted with existing state-of-the-art methods that have addressed the subject variability issue.

**Questions:**

The proposed model, which simply combines existing HyperNetwork and EEGNet, lacks substantial innovation. In addition, it is difficult to confirm the advantages of the proposed model from comparative experiment results as the performance improvement of the proposed method has not been consistently demonstrated across multiple datasets. This is, the proposed model achieved improved performance on the Dreyer et al. dataset in Table 1, while its performance degraded on the BCI Competition IV IIa dataset in Table 2. Furthermore, there has been no meaningful discussion about these conflicting results.

---

> ### Author Response · Authors · 2024-11-28
>
> We thank the reviewer for sharing their opinions and suggestions.
>
> We also thank the reviewer for acknowledging the motivation behind the work addressing cross-user variability and BCI illiteracy in the motor imagery domain.
> However, we would like to draw the reviewer’s attention towards the approach of using resting state data as a means to learn user-specific representations. Previous literature (discussed in Introduction section on Page 2, 59-64) cites specific markers in resting state data, which highly correlates with BCI illiteracy and indicates the relevant user-specific information in resting state data when performing motor imagery. These works motivate our work specifically to use resting state data and not just the motor imagery data recorded during the experiments.
> Based on the reviews, we have updated the Appendix Section in the original submission and the updated document can be viewed in the submission.
>
> Following are the point-wise responses to the reviewer’s comments:
>
> _There is no substantial innovation in proposed method combining the conventional HyperNetworks and EEGNet._
>
> As far as we know, this work is the first attempt to include resting state data to learn data-driven representations for motor imagery tasks. Previous works have used markers from resting state to predict the extent of BCI illiteracy.
> It would be helpful if the reviewer could direct us to relevant sources they found similar to the proposed method. We agree that the concepts of Hypernetworks and EEGNet are not novel, but their application in the current context by learning EEGNet weights for the downstream task of motor imagery using resting state is unexplored and novel.
>
>
> _The performance improvement of the proposed method over existing EEGNet has not been consistently demonstrated across multiple datasets. This is, the proposed model achieved improved performance on the Dreyer et al. dataset, while its performance degraded on the BCI Competition IV IIa dataset._
>
>
> We acknowledge the reviewer’s concern. The motivation here was to demonstrate the current approach's relevance on two different dataset sizes, where BCI competition dataset has fewer participants than Dreyer et al.
> We also perform Leave-N-Out evaluation with varying combinations on large datasets like Dreyer et al. to verify the effect of training dataset size.
>
> _Furthermore, there has been no meaningful discussion about these conflicting results._
>
> We would like to draw attention to section 4.1, page 6, 283-285 of the paper, where we discuss how comparison encourages to collection of larger datasets and the inherent assumption of the conflicting results being a smaller number of participants.
> Based on the reviews, we perform the following additional evaluation on existing and several other MI datasets:
> 1. Use statistical tests to confirm significance.
> 2. Perform Leave N out with N values ranging from 8-32 to validate the effect of training dataset size.
>
> _No comparisons were conducted with existing state-of-the-art methods that have addressed the subject variability issue._
>
> Several architectures have been proposed to address subject variability issues, however the approaches have focused on transfer learning or few shot paradigms that use labelled data from target participants.
>
> The following works have used a subject-independent / zero-shot / leave-one-subject-out strategy to test the effectiveness without training on labelled data from the target subject. However, the approaches and their performance depend on the architecture used for the dataset. While our aim is to test the effectiveness of using resting state information for downstream EEG classification. It is preferable to evaluate considering the baseline with the main architecture (EEGNet) for task classification. We choose EEGNet as it has been well interpreted and benchmarked across various paradigms in EEG domain.
>
> O. -Y. Kwon, M. -H. Lee, C. Guan and S. -W. Lee, "Subject-Independent Brain–Computer Interfaces Based on Deep Convolutional Neural Networks," in IEEE Transactions on Neural Networks and Learning Systems, vol. 31, no. 10, pp. 3839-3852, Oct. 2020, doi: 10.1109/TNNLS.2019.2946869.
>
> Zhang, Kaishuo, et al. "Adaptive transfer learning for EEG motor imagery classification with deep convolutional neural network." Neural Networks 136 (2021): 1-10.
>
> Following paper uses domain adaptation, and has evaluated their effectiveness on BCI Competition IVa and IVb datasets. However, domain adaptation/semi-supervised approaches are not mutually exclusive and may be integrated with our approach to optimize the performance.
>
>
> Zhang, Dongxue, et al. "MI-DAGSC: A domain adaptation approach incorporating comprehensive information from MI-EEG signals." Neural Networks 167 (2023): 183-198.

---

> ### Author Response · Authors · 2024-12-02
>
> Dear Reviewer gs65
>
> We kindly follow up on your feedback on our manuscript since today is the last discussion day.
>
> In our earlier responses, we believe we have addressed your concerns comprehensively. We are eager to know if there are any additional suggestions or specific points we could consider to enhance our manuscript further.
>
> We sincerely hope you might provide us with further insights that could guide us in strengthening our work.
>
> Thank you for your time and consideration.

---

### Meta-Review · Area_Chair_uBa4 · 2024-12-17

**Metareview:**

This paper presents a novel architecture, HyperEEGNet, which integrates HyperNetworks (HNs) with the EEGNet architecture, to improve EEG-based BCI systems, addressing the limitations of long calibration sessions and the underutilization of resting-state EEG data in motor imagery decoding tasks. Tacking variability across subjects is an important issue in BCI as well. The paper demonstrates that the proposed model  has domain generalization capabilities. There are a few concerns raised by reviewers. The main criticism is in limited dataset size and performance evaluation. It will be better to use more diverse larger datasets for performance evaluation. Since there are a lot of work on handling subject variability in BCI, it will be better to include some comparisons with other approaches. Therefore, the paper is not recommended for acceptance in its current form. I hope authors found the review comments informative and can improve their paper by addressing these carefully in future submissions.

**Additional Comments On Reviewer Discussion:**

While the authors made efforts for rebuttal, there was no change during the discussion period. All of reviewers stood by their original decisions.

---

### Decision · Program_Chairs · 2025-01-22

Reject